physical chemistry

Gemini quaternary ammonium salt surfactant, carboxylic counterions, surface properties, micellization, bacteriostatic activity

**Authors for correspondence:**
Hongqin Liu
e-mail: liuhongqin@th.btbu.edu.cn
Baocai Xu
e-mail: xubaoc@btbu.edu.cn

This article has been edited by the Royal Society of Chemistry, including the commissioning, peer review process and editorial aspects up to the point of acceptance.

# The surface adsorption, aggregate structure and antibacterial activity of Gemini quaternary ammonium surfactants with carboxylic counterions

Xiqin Zhou, Siqi Hu, Yu Wang, Sana Ullah, Jun Hu, Hongqin Liu and Baocai Xu

School of Food and Chemical Engineering, Beijing Key Laboratory of Flavor Chemistry, Beijing Higher Institution Engineering Research Center of Food Additives and Ingredients, Beijing Technology and Business University, No. 11 Fucheng Road, Beijing 100048, People's Republic of China

(iD) HL, 0000-0003-2051-0342

A group of Gemini quaternary ammonium surfactants with the formula $C_nH_{2n+1}CONH(CH_2)_2N^+(CH_3)_2(CH_2)_2N^+(CH_3)_2(CH_2)_2NHCOC_nH_{2n+1} \cdot 2Y$ ($n$ = 11, 13 and 15, Y = $HCOO^-$, $CH_3COO^-$ and $CH_3CHOHCOO^-$) have been synthesized by a counterion conversion process and characterized by Fourier transform infrared spectroscopy and mass spectroscopy. Their adsorption and self-aggregation properties are investigated by surface tension, conductivity, dynamic light scattering and transmission electron microscopy (TEM) measurements. The results show that these surfactants reduce the surface tension of water to a minimum value of 26.51 mN m$^{-1}$ at a concentration of $5.72 \times 10^{-2}$ mmol l$^{-1}$. Furthermore, the increased alkyl chain length of the carboxylic counterions leads to the increased critical micelle concentration, the decreased degree of counterion binding ($\beta$) and the decreased self-assembly tendency, but the minimum area per surfactant molecule ($A_{min}$) adsorbed at the air–aqueous solution are similar. TEM images reveal that these surfactants self-assemble spontaneously into aggregates with vesicle or bilayer structures. It is also found that they have superior antibacterial activity at a concentration of 0.1 g l$^{-1}$. The high surface activity and high antibacterial activity of the Gemini quaternary ammonium salt surfactants containing different carboxylic counterions bring more possibilities for the application in the field of biomedicine.

# 1. Introduction

Ionic surfactants are widely used in chemistry, material science, biochemistry and life science due to their unique interfacial adsorption and self-assembly behaviour. Among them, anionic surfactants are often used in thickening, emulsification, decontamination and other aspects of research [1,2]. Therefore, more and more novel surfactant systems and their applications have been reported [3]. Compared with other types of surfactants, cationic surfactants have two significant characteristics: they are highly bactericidal and easily adsorbed on the general solid surface. Quaternary ammonium surfactants, as a kind of important cationic surfactant, are usually used in antistatic, corrosion inhibition, sterilization and other aspects due to their molecular structure of positive charge head group [4–6]. Meanwhile, cationic surfactant–DNA particles are frequently used in the biochemical research and have promising prospects for application in the field of biochemistry [7–9].

Gemini quaternary ammonium surfactant, which consists of two hydrophobic chains and two polar head groups covalently linked by a spacer, offers structural versatility and superior properties such as low critical micelle concentration (CMC) and high surface reduction efficiency when compared with quaternary ammonium monomer [10,11]. Focusing on their excellent properties, scientists worldwide have designed and synthesized some Gemini surfactants with different structures in recent decades. Some fundamental research on the interfacial adsorption behaviour and aggregation behaviour in aqueous solution has been reported [11–14]. Studies have shown that the main factors, which affect interfacial adsorption behaviour and micelle aggregation behaviour, are hydrophobic alkyl chain length, the spacer group, the polarity of the head group, the characteristics of counter ions, etc. [15,16]. But so far, few studies have reported the influence of negatively charged counterions on the aqueous surfactant solution. Kumari & Sundar *et al.* studied the effect of a Hofmeister series of inorganic counterions and *p*-toluenesulfonate on the solvation dynamics and rotational relaxation in aqueous micelles of hexadecyltrimethylammonium surfactants, and indicated the positive significance of this research on the physico-chemical properties of surfactants in aqueous solution [17]. At present, the research on the effect of positively charged counterions on surfactants has been reported and their mutual interactions from the microscopic mechanism have been discussed, but the effect of negatively charged organic counterions on quaternary ammonium surfactants has rarely been reported [18,19]. Secondly, in the present synthesis method, the anions of these surfactants are mostly inorganic halogen atoms, which affect their properties and limit their application to a certain extent [11]. Under the requirements of green chemistry, the green synthesis method and halogen-free counterions system are the development trends for gemini quaternary ammonium surfactants.

In this paper, a series of Gemini quaternary ammonium surfactants with the carboxylic counterions are synthesized and their surface adsorption, micelle formation and antibacterial activity are studied.

# 2. Materials and methods

## 2.1. Materials

Methyl laurate (99%), methyl myristate (98%), methyl palmitate (99.5%), methyl stearate (97%), triethylenetetramine (60%) and dimethyl carbonate (99.5%) were purchased from J&K Scientific Ltd. Toluene (99.5%), sodium hydroxide (85%), formic acid (98%), acetic acid (99.5%), lactic acid (85%), absolute ethanol (99.7%), absolute methanol (99.5%) and absolute ethyl ether (99.7%) were provided by Sinopharm Chemical Reagent Beijing Co., Ltd. Acetonitrile (99%) was obtained from an Beijing HWRK Chemical Reagent Co., Ltd. Nutrient broth was supplied by Beijing OBO Biotech Co., Ltd. Benzalkonium chloride solution (50 g l$^{-1}$) was purchased from Beijing Derun Pharmaceutical Co., Ltd. *Pseudomonas aeruginosa*, *Escherichia coli*, *Staphylococcus aureus* and *Bacillus subtilis* were obtained from an environmental professional laboratory. The high-purity water ($\rho = 18.25\,M\Omega\,cm^{-1}$) was supplied by an ultrapure laboratory water purification system. *N,N,N′,N′*-tetramethyl-*N,N′*-di(2-laurylamideethyl) ethylenediamine methyl carbonate, *N,N,N′,N′*-tetramethyl-*N,N′*-di(2-myristylamideethyl) ethylenediamine methyl carbonate and *N,N,N′,N′*-tetramethyl-*N, N′*-di(2-palmitylamideethyl) ethylenediamine methyl carbonate (purity greater than 98%) were prepared according to a green method reported in our previous paper [20].

## 2.2. Synthesis of Gemini surfactants (*n*-2-*n*-2Y)

An amount of 0.1 mol *N,N,N,N*-tetramethyl-*N, N*-di (2-alkylamideethyl) ethylene diamine methyl carbonate (alkyl = lauryl, myristyl, palmityl) and moderate solvent methanol were introduced into a

**Figure 1.** Chemical structure of n-2-n-2Y.

250 ml round-bottom flask and stirred with 0.206 mol acid (formic acid, acetic acid and lactic acid) being added dropwise slowly. The reaction mixture was stirred for 4 h at 30–40°C. The crude products, which were obtained after methanol was removed, were recrystallized from acetonitrile to give the quaternary ammonium Gemini surfactants, the chemical structure of which is shown in figure 1. The yield values were 90.5–92.4%.

## 2.3. Confirmation of structure

An AB SCIEX API3200 LC/MS spectrometer and Thermo Fisher Nicolet ISLO FTIR spectrometer were used for mass spectroscopy (MS) analysis and Fourier transform infrared (FTIR) spectra records, respectively.

## 2.4. Measurement of Krafft temperature ($T_k$)

One per cent aqueous solution of the surfactants was prepared and kept in the refrigerator for 10 h at about 7°C. Then, the precipitated system was taken out of the refrigerator and placed in a low-temperature thermostat bath. A DDS-307 conductivity meter (cell constant is 0.997 cm$^{-1}$) was used to measure the conductivity of this surfactant solution. Each temperature gradient was measured three times to determine the average value with the standard deviation less than 0.2 μS cm$^{-1}$. The initial temperature of the solution was controlled as 7 ± 0.1°C and the measurements were carried out in 1°C increment with constant stirring by a glass rod.

## 2.5. Measurement of surface tension

A dataphysics tensiometer DCAT11 was used for the measurement of the static surface tension of this surfactant solution by the Wilhelmy plate method, within a concentration range of $1 \times 10^{-7}$ to $1 \times 10^{-2}$ mol l$^{-1}$ at 25°C. The surface tension of the high-purity water ($\rho = 18.25$ MΩ cm$^{-1}$) was equal to 72 mN m$^{-1}$ at 25°C. The determined value of each sample was the average of triplicate measurements, and the standard deviation was less than 0.02 mN m$^{-1}$.

## 2.6. Measurement of conductivity

The conductivity was measured at 25°C as explained above 2.4 and each sample was analysed three times to determine the average value with the standard deviation less than 0.016 μS cm$^{-1}$.

## 2.7. Measurement of dynamic light scattering

A Malvern Zetasizer Nano ZS instrument was used to determine the size distributions of self-assembled aggregate at 25°C with a solid-state He–Ne laser (output power of 22 mW at $\lambda = 632.8$ nm) as a light source. Through a 0.45 μm filter paper of mixed cellulose acetate, samples with concentrations of 2 × CMC and $1 \times 10^{-3}$ mol l$^{-1}$ were introduced into the glass cuvette and equilibrated for 2 min before measurement. The scanning scattering angle was set as 173°. The average values were determined by triplicate measurements of samples, and the standard deviation was less than 0.002. Using the CONTIN method, the correlation function of scattering data analysis was measured to obtain the distribution of diffusion coefficients ($D$) of the solutes. Then the Stokes–Einstein equation $R_h = kT/6\pi\eta D$ was used to determine the apparent equivalent hydrodynamic radius ($R_h$), where $k$ was the Boltzmann constant, $T$ was the absolute temperature and $\eta$ was the solvent viscosity.

## 2.8. Transmission electron microscope

Using the Oxford X-MAX JEM-2100 with uranyl acetate solution (2%) as the staining agent, transmission electron microscopy (TEM) was measured by a negative-staining method. A drop of surfactant solution

with a concentration of $2 \times$ CMC and $1 \times 10^{-3}$ mol $l^{-1}$ was dropped onto the clean sealing film. A porous carbon support membrane (260 mesh) was put upside down on the surface of the droplet for 10 min. Then the droplet was sucked from the edge, and the porous carbon support membrane was dried for at least 12 h in a vacuum dryer. After that, a drop of 1% dioxy uranium acetate was added to another clean sealing film, and the adsorbed carbon support membrane was put upside down on the droplet. After 15 s of dyeing, the porous carbon support membrane was dried for at least 12 h in the vacuum dryer. Lastly, the samples were imaged at a working voltage of 120 kV.

## 2.9. Measurement of antibacterial activity

Antibacterial activities of these surfactants were determined by monitoring the growth of Gram-negative bacteria (*P. aeruginosa* and *E. coli*) and Gram-positive bacteria (*S. aureus* and *B. Subtilis*) using a Finland Bioscreen automatic growth curve instrument. 1.8 g nutrient broth powder and 100 ml distilled water were mixed to prepare the fluid nutrient broth and then sterilized by autoclave. Two hundred microlitres of bacteria were introduced to the fluid nutrient broth and cultivated for 12 h with oscillation at 37°C. Then, the freshly prepared sterilized fluid nutrient broth was added to the previous 200 µl suspension liquid and oscillated continuously for 6 h to completely motivate the bacteria.

Fifty microlitres of sterile water was mixed with the 450 µl blank nutrient medium to set as the baseline group, and the blank group consisted of 50 µl sterile water and 450 µl bacterial solution. The control group was prepared by mixing 50 µl Benzalkonium chloride solution with a concentration of 1.00 g $l^{-1}$ with 450 µl bacterial solution. Fifty microlitres of surfactant solutions with a concentration gradient of 0.10, 0.50 and 1.00 g $l^{-1}$ were prepared by sterile water and mixed with the 450 µl bacterial solution. The prepared solution was separately placed into the middle part of the 100-well plates and monitored for 24 h with oscillation at 37°C, and the optical density (OD) value at 600 nm was measured every other hour.

# 3. Results and discussion

## 3.1. Characterization of Gemini quaternary ammonium surfactants

The detailed characterization of all the surfactants was carried out by FTIR spectroscopy and an MS. Spectral characterizations results are shown in the electronic supplementary material. In brief, the structures of all these compounds were confirmed.

## 3.2. Krafft temperature ($T_k$)

The Krafft temperature of these surfactants is determined from the plot of the specific conductance ($\kappa$) versus temperature in the aqueous solution. The corresponding graphs are shown in electronic supplementary material, figure S2. Seen from electronic supplementary material, figure S2, in the relatively low temperature range, the conductance of the surfactant solution has a small change with the increase of the temperature, and then the sharp increase is observed within a relatively narrow temperature range that represents the sharp increase in surfactant solubility. The $T_k$ of the surfactants can be obtained from the inflection point of the $\kappa$–$T$ curve (table 1).

## 3.3. Surface activity and surface property parameters

Figure 2 shows the variation in surface tension versus the concentration of these surfactant aqueous solutions. The CMC values of all surfactants are determined by the inflection point of the $\gamma$–$c$ curve. Although there are still some debates [21,22], the Gibbs adsorption equations shown as follows are used to calculate the maximum surface excess concentration ($\Gamma_{max}$) and the minimum area per surfactant molecule ($A_{min}$) adsorbed at the air–aqueous solution interface [23],

$$\Gamma_{max} = \left(\frac{-1}{2.303nRT}\right)\left(\frac{d\gamma}{d\log c}\right)$$ (3.1)

and

$$A_{min} = \frac{1}{N_A \Gamma_{max}},$$ (3.2)

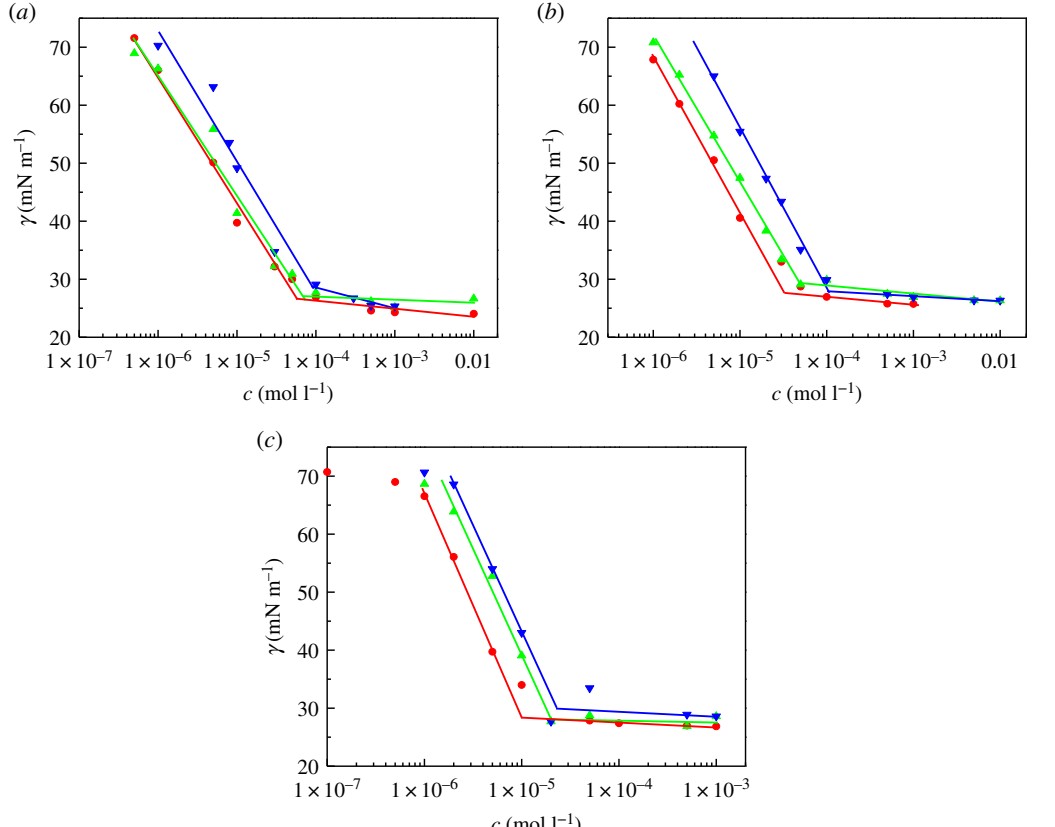

**Figure 2.** Plots of $\gamma$ against the concentration ($c$) of these surfactants at 25°C, ($a$) 11-2-11-2Y, ($b$) 13-2-13-2Y and ($c$) 15-2-15-2Y, Y = HCOO$^-$ (filled circle), CH$_3$COO$^-$ (filled triangle) and CH$_3$CHOHCOO$^-$ (inverted filled triangle).

**Table 1.** The $T_k$ of 11-2-11-2Y, 13-2-13-2Y, 15-2-15-2Y (Y = HCOO$^-$, CH$_3$COO$^-$ and CH$_3$CHOHCOO$^-$) aqueous solution.

| Gemini surfactants | $T_k$ (°C) |
| --- | --- |
| 11-2-11-2HCOO$^-$ | 17 |
| 11-2-11-2CH$_3$COO$^-$ | 18 |
| 11-2-11-2CH$_3$CHOHCOO$^-$ | 18 |
| 13-2-13-2HCOO$^-$ | 18 |
| 13-2-13-2CH$_3$COO$^-$ | 18 |
| 13-2-13-2CH$_3$CHOHCOO$^-$ | 17 |
| 15-2-15-2HCOO$^-$ | 18 |
| 15-2-15-2CH$_3$COO$^-$ | 18 |
| 15-2-15-2CH$_3$CHOHCOO$^-$ | 19 |

where $R$ is the gas constant, $N_A$ is Avogadro's number, $T$ is the absolute temperature and $n$ is a constant. For these Gemini quaternary ammonium surfactants, $n$ is generally determined as 2 or 3 [24,25]. The value of $n$ has no effect on the variation trend of the values of $\Gamma_{max}$ and $A_{min}$. However, it has been widely assumed that it is more appropriate to use a value of 2 for $n$, because some reports suggest that as $n$ is 3, the calculated $A_{min}$ is unacceptably large [26]. In this case, a value of 2 for $n$ is used to calculate. The pC$_{20}$ values calculated by $-\log C_{20}$ are used to describe the efficiency to reduce the surface tension of the solvent by 20 mN m$^{-1}$. The following equations put forward by Rosen's methodology are used to calculate the Gibbs energy of adsorption ($\Delta G_{ads}$) and the standard Gibbs energy of micellization ($\Delta G_m^\theta$) [27],

$$\Delta G_m^\theta = RT \ln\left(\frac{\text{CMC}}{55.5}\right) \tag{3.3}$$

and

$$\Delta G_{ads} = \Delta G_m^{\theta} - 6.023(\gamma_0 - \gamma_{CMC})A_{min},$$ (3.4)

where $\gamma_0$ is the surface tension of the ultrapure water. The above calculation results are listed in table 2.

As seen in table 2, the CMC and $\gamma_{CMC}$ values of these surfactants with the same length of hydrophobic chains increase with increasing the alkyl chain length of carboxylic counterions. This is because the hydrophilicity of the counterion is one of the primary effects determining micellization: the higher the hydrophilicity of a counterion, the more unfavourable micellization becomes, leading to high CMC [28]. The CMC values of the corresponding conventional quaternary ammonium surfactants with carboxylate anion [$CH_3(CH_2)_nCH_2N^+(CH_3)_3 \cdot A^-$, $n = 10, 12$, A = $HCOO^-$, $CH_3COO^-$] are two orders of magnitude higher than that of these Gemini surfactants, although the hydrophobic chain lengths are different [29]. Some Gemini quaternary ammonium surfactants with similar structures [$n$-2-$n$: $C_nH_{2n+1}CONH(CH_2)_2N^+(CH_2CH_3)_2(CH_2)_2N^+$ $(CH_2CH_3)_2(CH_2)_2NHCOC_nH_{2n+1} \cdot 2Br$, $n = 11, 13, 15$] have been reported [30]. Compared with the CMC values of 11-2-11, which is $1.0 \times 10^{-3}$ mol l$^{-1}$, the CMC values of 11-2-11-2Y are much lower; especially, the CMC values of 11-2-11-2HCOO− and 11-2-11-2CH$_3$COO$^-$ are two orders of magnitude lower. But, the CMC values of 13-2-13-2Y and 15-2-15-2Y are one order of magnitude lower compared with those of 13-2-13 and 15-2-15 ($7.5 \times 10^{-4}$ and $5.0 \times 10^{-4}$ mol l$^{-1}$). This indicates that although micellization of Gemini surfactants is affected by both the hydrophilicity of counterions and hydrocarbon parts of the polar headgroup [31], but with the increase of alkyl chain length, the main reason for the decrease of CMC is that the increase of alkyl chain length leads to the increase of hydrophobicity of surfactants.

Gao *et al*. [32] suggested that the hydrophobicity and the hydration radius of the organic counterions were two factors affecting the values of $A_{min}$ and $\Gamma_{max}$. But from table 2, it can be seen that these three species of carboxylic counterions have little effect on the values of $\Gamma_{max}$ and $A_{min}$. The reason may be that, for gemini quaternary ammonium salts with the short alkyl chain counterions, the dominating factors influencing the $\Gamma_{max}$ values are the size of hydrophilic head groups and the electrical repulsion between hydrophilic head groups [33]. In addition, counterions such as formic acid and acetate, which have strong hydrophilicity and weak hydrophobicity, will not or will seldom insert into the surface adsorption layer, thus the surface adsorption parameters of these quaternary ammonium salts have little change. Meanwhile, table 2 also shows that the pC$_{20}$ values of these surfactants with the same length of hydrophobic chains increase fractionally with decreasing the alkyl chain length of carboxylic counterions, which indicates that Gemini surfactants with formate counterion have higher efficiency in reducing the surface tension. Because of the negative $\Delta G_{ads}$ and $\Delta G_m^{\theta}$ values (table 2), it can be concluded that both adsorption and micellization of these surfactants are spontaneous at 25°C. Furthermore, the adsorption of these surfactant systems is probably preferential than micellization because of the more negative $\Delta G_{ads}$ values.

## 3.4. Conductivity measurement

Electronic supplementary material, figure S3 shows the conductivity values ($\kappa$) of 11-2-11-2Y, 13-2-13-2Y, 15-2-15-2Y aqueous solution as a function of their concentrations ($c$). It is noteworthy that some results of conductivity measurements (electronic supplementary material, figure S3 A-II, B-II, C-I and C-II) seem to deviate from linearity at concentrations lower than CMC. This phenomenon could be a consequence of ionic pairing or premicellar associations, which can be analysed by the shape of $\kappa$ versus $c$ and molar conductivity $\Lambda$ ($\Lambda = (\kappa - \kappa_0)/c$, $\kappa_0$ is the conductivity of water) versus $c^{0.5}$ plots at $c <$ CMC [34]. Ion pairing results in a neutralization of electrical charges and thus in a loss of conductivity, so the corresponding $\kappa$ versus $c$ and $\Lambda$ versus $c^{0.5}$ plots curve toward the concentration. On the other hand, premicellar association of ionic surfactant gives rise to small aggregates of surfactant ions, which results in an increase in the conductivity of the solution; therefore, the corresponding $\kappa$ versus $c$ plot will show a curvature toward the $\kappa$ axis. In addition, $\Lambda$ will increase with $c$. At higher concentration, where micelles appear in the solution and bind counterions, the value of $\Lambda$ will start decreasing. Thus, the plot of $\Lambda$ versus $c^{0.5}$ will present a maximum. So, according to electronic supplementary material, figure S3 A-II, B-II, C-I and C-II, and electronic supplementary material, figure S4, ion pairing may occur in solutions of 11-2-11-2CH$_3$COO$^-$, 13-2-13-2CH$_3$COO$^-$ and 15-2-15-2CH$_3$COO$^-$, and premicellar association may be present for 15-2-15-2HCOO$^-$.

The CMC of the surfactants can be obtained from the inflection point of the $\kappa$-$c$ curve. The ionization degree of micelle ($\alpha$) is obtained from the ratio of the slope of the linear portions right of the inflection

**Table 2.** Characteristic parameters of 11-2-11-2Y, 13-2-13-2Y, 15-2-15-2Y (Y = HCOO$^-$, CH$_3$COO$^-$ and CH$_3$CHOHCOO$^-$) aqueous solution at 25°C. The method 1 stands for surface tension measurement. The method 2 stands for conductivity measurement. The standard error $\sigma_\mu$ are $\sigma_\mu$(CMC) = 1 × 10$^{-3}$ mmol l$^{-1}$, $\sigma_\mu(\gamma_{CMC})$ = 0.01 mN m$^{-1}$, $\sigma_\mu(\gamma_{max})$ = 0.01 × 10$^{-10}$ mol cm$^{-2}$, $\sigma_\mu$(pC$_{20}$) = 0.01, $\sigma_\mu(A_{min})$ = 0.01 nm$^2$, $\sigma_\mu(A_{min})$ = 0.01 nm$^2$, $\sigma_\mu(A_{min})$ = 0.01 nm$^2$, $\sigma_\mu(\Delta G_{ads})$ = −1 kJ mol$^{-1}$ and $\sigma_\mu(\Delta G_{ads})$ = −1 kJ mol$^{-1}$.

| Gemini surfactant | CMC (mmol l$^{-1}$) | | $\gamma_{CMC}$ (mN m$^{-1}$) | pC$_{20}$ | $\Gamma_{max}$ (×10$^{-10}$ mol cm$^{-2}$) | $A_{min}$ (nm$^2$) | $\Delta G_{ads}$ (kJ mol$^{-1}$) | $\Delta G_m^{\theta}$ (kJ mol$^{-1}$) | | $\beta$ |
| --- | --- | --- | --- | --- | --- | --- | --- | --- | --- | --- |
| | method 1[a] | method 2[b] | | | | | | method 1[a] | method 2[b] | |
| 11-2-11-2HCOO$^-$ | 5.72 × 10$^{-2}$ | 5.73 × 10$^{-2}$ | 26.51 | 5.42 | 2.09 | 0.79 | −55.81 | −34.17 | −26.43 | 0.556 |
| 11-2-11-2CH$_3$COO$^-$ | 6.32 × 10$^{-2}$ | 6.18 × 10$^{-2}$ | 26.92 | 5.37 | 1.91 | 0.87 | −57.54 | −33.92 | −26.06 | 0.549 |
| 11-2-11-2CH$_3$CHOHCOO$^-$ | 0.101 | 0.105 | 27.63 | 5.14 | 1.90 | 0.88 | −56.28 | −32.76 | −23.84 | 0.512 |
| 13-2-13-2HCOO$^-$ | 3.16 × 10$^{-2}$ | 3.18 × 10$^{-2}$ | 27.62 | 5.40 | 2.41 | 0.69 | −54.08 | −35.64 | −26.76 | 0.509 |
| 13-2-13-2CH$_3$COO$^-$ | 4.09 × 10$^{-2}$ | 4.10 × 10$^{-2}$ | 27.99 | 5.20 | 2.69 | 0.62 | −51.43 | −35.00 | −25.70 | 0.492 |
| 13-2-13-2CH$_3$CHOHCOO$^-$ | 8.92 × 10$^{-2}$ | 6.78 × 10$^{-2}$ | 28.49 | 4.87 | 2.54 | 0.65 | −50.10 | −33.07 | −23.53 | 0.453 |
| 15-2-15-2HCOO$^-$ | 1.05 × 10$^{-2}$ | 1.41 × 10$^{-2}$ | 28.32 | 5.61 | 3.35 | 0.50 | −51.52 | −38.37 | −27.49 | 0.462 |
| 15-2-15-2CH$_3$COO$^-$ | 2.09 × 10$^{-2}$ | 2.19 × 10$^{-2}$ | 28.35 | 5.34 | 3.14 | 0.53 | −50.60 | −36.67 | −25.86 | 0.440 |
| 15-2-15-2CH$_3$CHOHCOO$^-$ | 2.63 × 10$^{-2}$ | 3.34 × 10$^{-2}$ | 29.09 | 5.23 | 3.09 | 0.54 | −50.06 | −36.10 | −24.93 | 0.442 |

point at CMC versus left. The relation: $\beta = 1 - \alpha$ is used to calculate the degree of counterion binding ($\beta$) [35]. $\Delta G_m^\theta$ is calculated according to the following equation [36]:

$$\Delta G_m^\theta = RT \left( \frac{1}{2} + \beta \right) \ln \text{CMC} - \frac{1}{2} RT \ln 2. \qquad (3.5)$$

The values of CMC, $\beta$ and $\Delta G_m^\theta$ are shown in table 2. There is little or no distinction between the CMC values measured by conductivity measurements and by surfactant tension measurements. Table 2 shows that the variation trend of $\beta$ values of ammonium formate, ammonium acetate and ammonium lactate with the same length of hydrophobic chains is gradually decreasing. The reason is probably that, for the micelles of the same size and structure, the larger the hydration radius of the counterion, the lower the degree of counterion binding [23]. Simultaneously, for quaternary ammonium surfactants, the increase of the degree of counterion binding will lead to the decrease of the CMC of the surfactants in aqueous solution. This inference accords with the experimental results that the CMC gradually increases with increasing the chain length of the carboxylic counterions. Table 2 also shows that there is some difference between $\Delta G_m^\theta$ values obtained by the two methods, but they have the same rule of change. Moreover, when the counterions are in turn $HCOO^-$, $CH_3COO^-$ and $CH_3CHOHCOO^-$, the absolute value of the $\Delta G_m^\theta$ values gradually decreases for these surfactants with the same length of hydrophobic chains, indicating that Gemini surfactants with formate counterion are most conductive to forming micelles. This is consistent with the measurement results of CMC.

## 3.5. Dynamic light scattering and TEM measurements

The dynamic light scattering (DLS) data and TEM micrographs of surfactant solutions at a concentration of $2 \times$ CMC are frequently used to evaluate their molecular self-assembly behaviour. In the antibacterial activity subsequently discussed, at this concentration (about $0.05 \text{ g l}^{-1}$), the bacteriostatic efficiency of these Gemini surfactant aqueous solutions was 53.9–80.2% for Gram-positive bacteria and Gram-negative bacteria. However, at the concentration of $1 \times 10^{-3} \text{ mol l}^{-1}$ (about $0.1 \text{ g l}^{-1}$), the antibacterial efficiency of all these surfactant solutions is above 75.3%, most of which are 80–95%. So, the self-assembly behaviour of these Gemini surfactants aqueous solutions at higher concentration ($1 \times 10^{-3} \text{ mol l}^{-1}$) was also investigated.

Seen from electronic supplementary material, figure S5, the $R_h$ values of these surfactant solutions with the concentration of $2 \times$ CMC are distributed in the range of 40–2000 nm, while the $R_h$ values of $1 \times 10^{-3} \text{ mol l}^{-1}$ are distributed in the range of 10–3000 nm. It is suggested that the increasing concentration of these surfactant solutions leads to more diverse and larger size aggregates. At the concentration of $2 \times$ CMC, two peaks are shown in the surfactants with formate counterion in the case of the same hydrophobic chain length. The first peak is about tens of nanometres although the intensity of 13-2-13-2HCOO$^-$ is not obvious, and the other one is generally hundreds of nanometres. The result indicates that the surfactants with formate counterion tend to form smaller aggregates at the low concentration. At the concentration of $1 \times 10^{-3} \text{ mol l}^{-1}$, at least two hydrated radius distribution peaks were observed. The peak of tens of nanometres may correspond to microvesicles, and the peak of hundreds of nanometres may be assigned to more complex aggregates, such as multilamellar vesicles (MLV), tubular micelle and lamellar micelle. Accordingly, the morphology of aggregates of these Gemini surfactants in aqueous solution with the concentration of $2 \times$ CMC and $1 \times 10^{-3} \text{ mol l}^{-1}$ has been carried out on TEM measurement, which is shown in figures 3–5.

Seen from figures 3–5, the size of these aggregates is practically consistent with the results of DLS. In the series of 11-2-11-2Y, at the concentration of $2 \times$ CMC, vesicles with a diameter of about 300 nm are formed in the aqueous solution of 11-2-11-2HCOO$^-$ and 11-2-11-2CH$_3$COO$^-$ (figure 3$a$(i)(ii)), while at the concentration of $1 \times 10^{-3} \text{ mol l}^{-1}$, MLV with a diameter of about 500 nm are formed (figure 3$b$(i)(ii)). In addition, at the concentration of $2 \times$ CMC, a mass of hollow slender bilayer micelles are observed in the aqueous solution of 11-2-11-2CH$_3$CHOHCOO$^-$ (figure 3$a$(iii)), while at the concentration of $1 \times 10^{-3} \text{ mol l}^{-1}$, vesicles are formed (figure 3$b$(iii)). This indicates that the introduction of −OH and the increase in the length of the alkyl carbon chain in the counterions cause the deformation of the aggregates at the concentration of $2 \times$ CMC, but the effect of counterions seems to be insignificant at relatively high concentration. This trend is also present in the series of 13-2-13-2Y. The mixed micelles of vesicles and deformed bilayers structure are observed in figure 4$a$(i), while uniform short tubular micelles are observed in figure 4$a$(ii)(iii). However, hollow and longish tubular

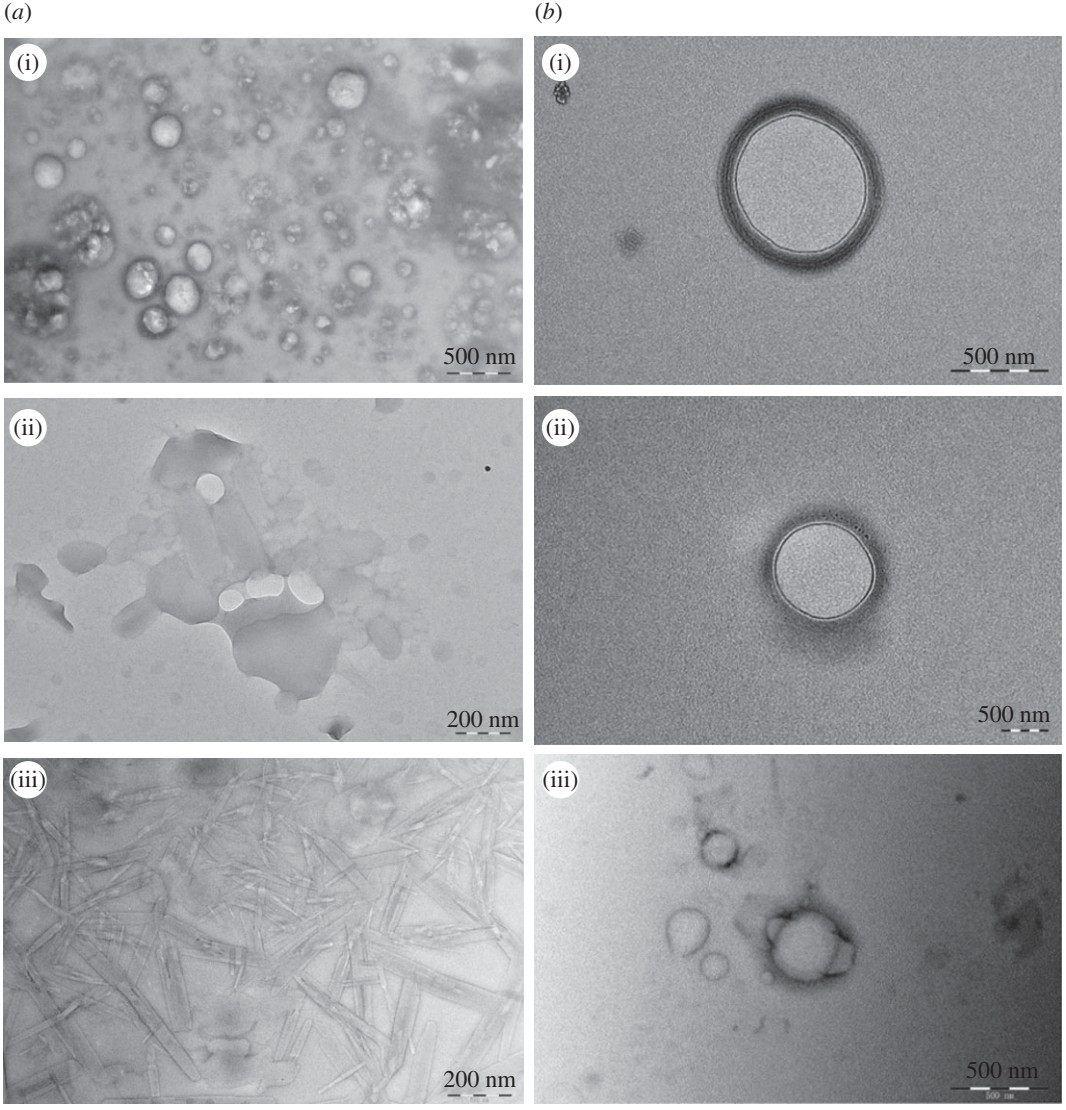

**Figure 3.** TEM micrographs of 11-2-11-2Y in aqueous solution of the concentration of $2 \times$ CMC (*a*) and $1 \times 10^{-3}$ mol l$^{-1}$ (*b*), (i) Y = HCOO$^-$, (ii) CH$_3$COO$^-$ and (iii) CH$_3$CHOHCOO.

micelles are all observed among that at a high concentration (figure 4*b*(i)(ii)(iii)). In spite of that, in the series of 15-2-15-2Y, the counterions have little influence on the self-assembly behaviour of these surfactants whether at a low or high concentration. Seen from figure 5, the vesicles with a diameter of about 100–200 nm are formed at the concentration of $2 \times$ CMC, while mixed micelles of vesicle and longish tubular structure are formed at the concentration of $1 \times 10^{-3}$ mol l$^{-1}$. The formation of various micelles is mainly due to the self-organization of quaternary ammonium cation driven by entropy. So, one of the major factors leading to the differences may be the distinction of hydrophobic chain length of these systems.

## 3.6. Packing parameter (P) of the aggregates structure

The hydrophilic head groups of aggregates formed by the surfactant molecules pack towards the water phase and the hydrophobic tail chains are away from the water phase in aqueous solution. Due to the different volumes of head groups and the tail chains, the aggregates have a diverse morphology. According to the space ($V_H$) occupied by the hydrophilic group and the chain length ($l_c$) of the hydrophobic group, the *P*-values are calculated by the following equation [37]:

$$P = \frac{V_H}{l_c a_0},$$
(3.6)

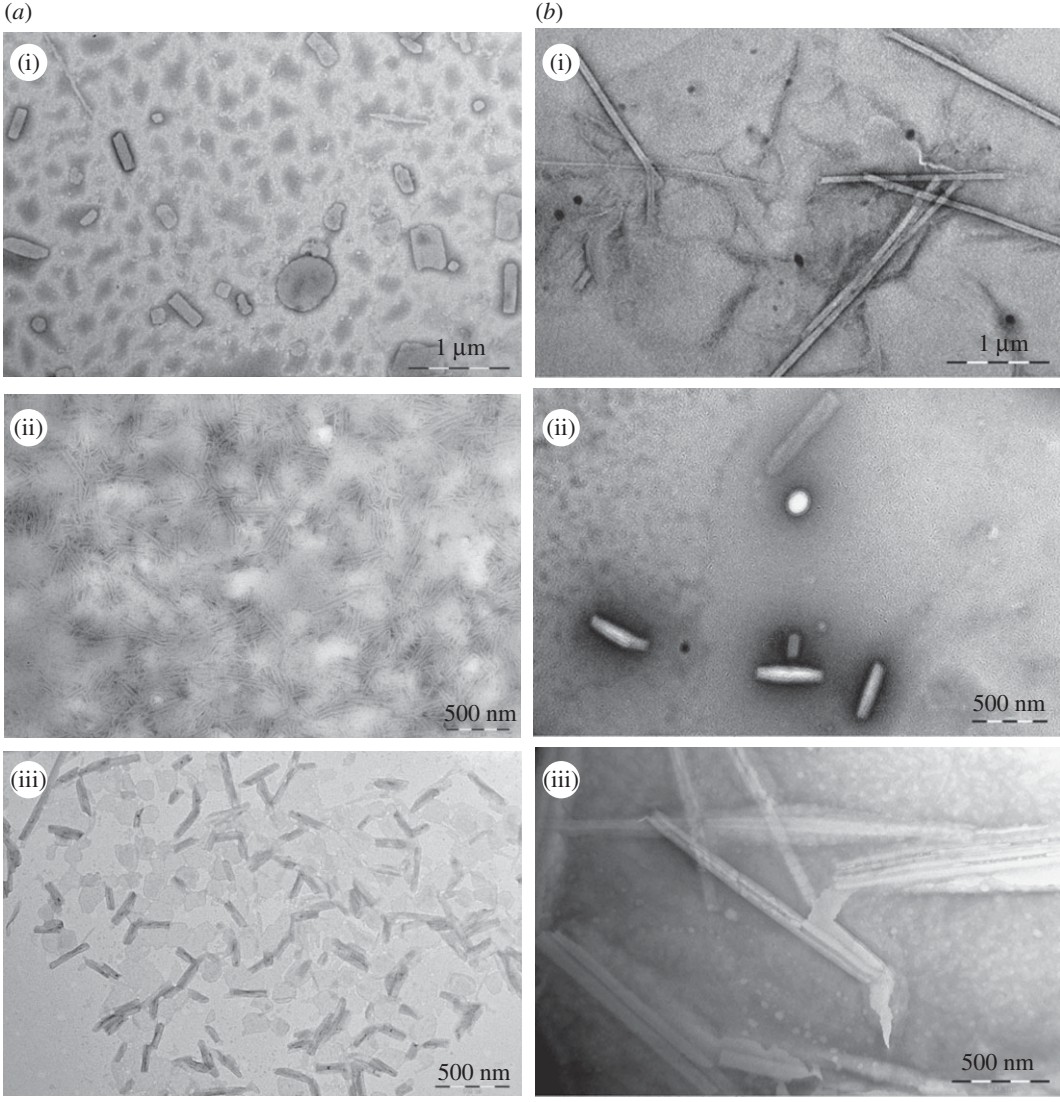

**Figure 4.** TEM micrographs of 13-2-13-2Y in aqueous solution of the concentration of $2 \times$ CMC (*a*) and $1 \times 10^{-3}$ mol $l^{-1}$ (*b*), (i) Y = HCOO$^-$, (ii) CH$_3$COO$^-$ and (iii) CH$_3$CHOHCOO$^-$.

where $a_0$ can be approximated to a half of $A_{min}$ with $n = 2$ [38], and the Tanford equation is used to calculate the $V_H$ and $l_c$ values [20,37],

$$V_H = 0.0274 + 0.0269\,n \tag{3.7}$$

and

$$l_c \leq 0.15 + 0.1265n, \tag{3.8}$$

where $n$ is the hydrophobic alkyl chain length inserted into the aggregates, $l_c$ is approximated to 80% of the calculated value for the fully extended alkyl chains. The values of $P$, which are between $1/2$ and 1, are listed in table 3. Combining with the relationship between $P$ and the structures of aggregates [20,37], when $1/2 < P < 1$, the geometry of surfactants turns into an ellipse cone with wide hydrophilic head base area and tiny hydrophobic chain and it forms a flexible and closed bilayer structure more readily. So, it is reasonable for these Gemini surfactants to form vesicles and bilayer aggregates observed by TEM.

## 3.7. Measurements of antibacterial activity

As a representation, the plots of the OD values of *P. aeruginosa* containing different concentrations of 11-2-11-2Y versus time (*h*) are shown in electronic supplementary material, figure S6. From electronic supplementary material, figure S6, it can be seen that the OD values of the blank rise over time

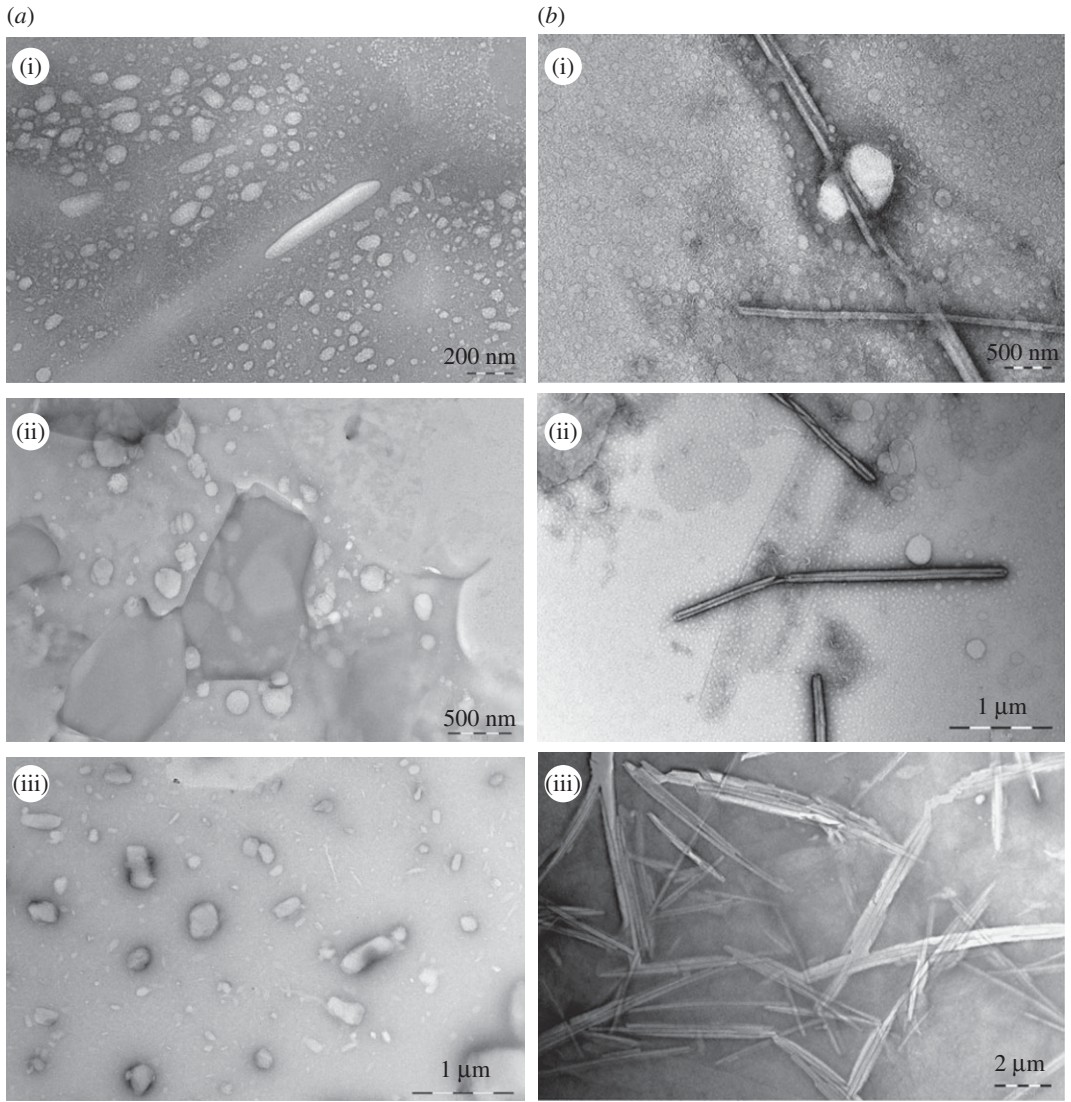

**Figure 5.** TEM micrographs of 15-2-15-2Y in aqueous solution of the concentration of $2 \times CMC$ (*a*) and $1 \times 10^{-3}$ mol l$^{-1}$ (*b*), (i) Y = HCOO$^-$, (ii) CH$_3$COO$^-$ and (iii) CH$_3$CHOHCOO$^-$.

**Table 3.** Values of *P* of 11-2-11-2Y, 13-2-13-2Y, 15-2-15-2Y (Y = HCOO$^-$, CH$_3$COO$^-$ and CH$_3$CHOHCOO$^-$).

| Gemini surfactants | $a_0$(nm$^2$) | *P* |
|---|---|---|
| 11-2-11-2HCOO$^-$ | 0.395 | 0.66 |
| 11-2-11-2CH$_3$COO$^-$ | 0.435 | 0.60 |
| 11-2-11-2CH$_3$CHOHCOO$^-$ | 0.44 | 0.60 |
| 13-2-13-2HCOO$^-$ | 0.345 | 0.76 |
| 13-2-13-2CH$_3$COO$^-$ | 0.31 | 0.85 |
| 13-2-13-2CH$_3$CHOHCOO$^-$ | 0.325 | 0.81 |
| 15-2-15-2HCOO$^-$ | 0.25 | 1.05 |
| 15-2-15-2CH$_3$COO$^-$ | 0.265 | 0.99 |
| 15-2-15-2CH$_3$CHOHCOO$^-$ | 0.27 | 0.97 |

initially. After 18 h, OD values achieved stability, which indicates that *P. aeruginosa* enters a stable status. After 30 h, the OD values decrease slightly, which indicates that *P. aeruginosa* enters the decay period. The same tendencies are observed in the groups containing 11-2-11-2Y, merely their OD values are smaller

**Table 4.** The bacteriostatic efficiency of 11-2-11-2Y, 13-2-13-2Y, 15-2-15-2Y (Y = HCOO⁻, CH₃COO⁻ and CH₃CHOHCOO⁻) at different concentration (g l⁻¹) at 37°C.

| Gemini surfactant | P. aeruginosa (%) | | | E. coli (%) | | | S. aureus (%) | | | B. subtilis (%) | | |
|---|---|---|---|---|---|---|---|---|---|---|---|---|
| | 0.01 | 0.05 | 0.1 | 0.01 | 0.05 | 0.1 | 0.01 | 0.05 | 0.1 | 0.01 | 0.05 | 0.1 |
| 11-2-11-2HCOO⁻ | 62.7 | 78.4 | 91.6 | 64.3 | 86.4 | 91.3 | 77.2 | 80.9 | 93.2 | 64.3 | 86.4 | 91.3 |
| 11-2-11-2CH₃COO⁻ | 60.7 | 68.5 | 83.1 | 60.3 | 82.8 | 86.7 | 75.7 | 80.0 | 92.3 | 60.3 | 82.8 | 86.7 |
| 11-2-11-2CH₃CHOHCOO⁻ | 54.9 | 67.2 | 75.3 | 53.9 | 74.2 | 82.7 | 75.1 | 79.3 | 89.3 | 53.9 | 74.2 | 82.7 |
| 13-2-13-2HCOO⁻ | 63.0 | 80.2 | 92.0 | 66.8 | 87.3 | 92.9 | 78.4 | 82.4 | 93.8 | 66.8 | 87.3 | 92.9 |
| 13-2-13-2CH₃COO⁻ | 62.5 | 71.9 | 87.3 | 62.7 | 87.0 | 90.9 | 77.4 | 81.9 | 93.4 | 62.7 | 87.0 | 90.9 |
| 13-2-13-2CH₃CHOHCOO⁻ | 58.9 | 67.0 | 77.6 | 54.5 | 82.0 | 89.5 | 76.3 | 80.8 | 92.8 | 54.5 | 82.0 | 89.5 |
| 15-2-15-2HCOO⁻ | 76.0 | 87.7 | 92.9 | 68.8 | 89.6 | 93.5 | 80.2 | 84.4 | 94.7 | 68.8 | 89.6 | 93.5 |
| 15-2-15-2CH₃COO⁻ | 63.0 | 84.1 | 91.2 | 61.6 | 88.6 | 91.9 | 79.4 | 84.5 | 94.7 | 61.6 | 88.6 | 91.9 |
| 15-2-15-2CH₃CHOHCOO⁻ | 59.1 | 76.6 | 81.5 | 60.9 | 82.7 | 90.5 | 77.9 | 83.3 | 94.6 | 60.9 | 82.7 | 90.5 |
| benzalkonium chloride solution | \ | \ | 76.0 | \ | \ | 74.0 | \ | \ | 79.0 | \ | \ | 74.0 |

than that of the blank, suggesting the antibacterial ability of these Gemini surfactants. In addition, the OD values decrease with the increasing concentration of the same surfactant. The bacteriostatic efficiency is calculated according to the following formula:

$$X = \frac{(OD_0 - OD_1)}{OD_0} \times 100\%, \tag{3.9}$$

where $OD_0$ and $OD_1$ refer to the OD values of the bacillus without surfactants and with surfactants at the same time, respectively. The values at 15 h are used to evaluate the surfactant inhibition for *S. aureus* and *S. aureus*, and the values at 10 and 12 h are, respectively, used to evaluate for *E. coli* and *B. subtilis*, which are listed in table 4. As a control, the antibacterial efficiency of $0.1\,\mathrm{g\,l^{-1}}$ benzalkonium chloride solution is also listed in table 4.

Table 4 shows that, at the concentration of $0.05\,\mathrm{g\,l^{-1}}$, the antibacterial activities of most of these Gemini surfactants are better than those of $0.1\,\mathrm{g\,l^{-1}}$ benzalkonium chloride. At the concentration of $0.1\,\mathrm{g\,l^{-1}}$, all these surfactants have excellent bacteriostatic effect both on Gram-positive bacteria and Gram-negative bacteria. In addition, among these bacteria, the surfactant inhibition for *S. aureus* is best, followed by the *E. coli*. Table 4 also lists that, for these Gemini quaternary ammonium salt surfactants with the same counterions, the bacteriostatic efficiency increases with the increase of hydrophobic carbon chain. Combined with the diffusion double layer theory, bacteria with negative charge have electrostatic interaction with the positive charge micelles, and a long hydrophobic chain is one of its driving forces. Gemini surfactants with longer hydrophobic carbon chain lengths can penetrate and diffuse into the cell membrane of the bacteria through the surface more easily, hinder the semi-permeability of the cell membrane, and further reach the cell interior, deactivating enzymes, and failing to produce proteases, thus denaturing proteins and killing the bacteria cells. Moreover, the variation trend of the bacteriostatic effect of ammonium formate, ammonium acetate and ammonium lactate with the same length of hydrophobic chains is gradually decreasing. The reason may be that the counterions wrapped with the positive micelles are diffused, Gemini surfactants with shorter alkyl chain length of counterions have a larger diffusion region, which can prevent the bacteria from exchanging substances with the nutrient medium better.

# 4. Conclusion

To study the effects of carboxylate counterions on the properties of Gemini quaternary ammonium surfactants, nine surfactants with three carboxylic counterions are synthesized by a counterion conversion process. The current studies have shown that the CMC values of these surfactants are $1.05 \times 10^{-5}$ to $1.01 \times 10^{-4}\,\mathrm{mol\,l^{-1}}$, and the $\gamma_{CMC}$ values are $26.51-29.09\,\mathrm{mN\,m^{-1}}$. With an increase of the alkyl chain length of the carboxylic counterions, the CMC values increase, the values of $\beta$ and $pC_{20}$ and the absolute values of $\Delta G_m^\theta$ decrease, and $A_{min}$ values show little difference. This suggests that those Gemini surfactants with shorter alkyl chain length of the carboxylic counterions, to adsorb in the air–aqueous interface, form micelles more readily and have higher efficiency on reducing the surface tension, and those surfactants with longer alkyl chain length of the carboxylic counterions have larger hydration radius and weak combination degree. Meanwhile, the DLS results show that the increasing concentration of these surfactant solutions leads to more diverse and larger size aggregates. TEM results show that, in the series of 11-2-11-2Y and 13-2-13-2Y, the transformation of carboxylic counterions from formate to acetate to lactate induces the transition of aggregates from vesicles to slender bilayer micelles at a concentration of $2 \times CMC$. However, in the series of 15-2-15-2Y, such carboxylic counterions conversion had little effect on the morphology and size of the aggregates whether at a concentration of $2 \times CMC$ or $1 \times 10^{-3}\,\mathrm{mol\,l^{-1}}$. These Gemini surfactants are also found to have superior antibacterial activity at a concentration of $0.1\,\mathrm{g\,l^{-1}}$. The increasing alkyl chain length of the hydrophobic groups as well as the decreasing alkyl chain length of the carboxylic counterions leads to higher bacteriostatic efficiency. The high surface activity and high antibacterial activity of these Gemini quaternary ammonium salt surfactants containing different carboxylic counterions bring more possibilities for their application in the field of biomedicine.

Data accessibility. Our data are provided as electronic supplementary material.
Authors' contributions. J.H., H.L. and B.X. designed the research. X.Z. and J.H. performed the experiment. X.Z., J.H. and H.L. carried out the data analysis. X.Z. and H.L. drafted the manuscript. S.H., Y.W. and S.U. helped draft the paper. All authors gave final approval for publication.
Competing interests. We declare we have no competing interests.

Funding. The work was supported by National Key R&D Program of China (2017YFB0308701), the National Natural Science Foundation of China (21676003), Beijing Municipal Science and Technology Project (Z171100001317015), the Science and Technology Program Key Project of Beijing Municipal Commission of Education (KZ201510011010), the Transformation of Scientific and Technological Achievements-Promotion Plan Project (PXM2016_014213 000028).

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
