## [Reviewer comments · Royal Society Open Science]

Review History

RSOS-190378.R0 (Original submission)

Review form: Reviewer 1

Is the manuscript scientifically sound in its present form?

Yes

Are the interpretations and conclusions justified by the results?

Yes

Is the language acceptable?

Yes

Is it clear how to access all supporting data?

Yes

Do you have any ethical concerns with this paper?

No

Have you any concerns about statistical analyses in this paper?

Yes

Recommendation?

Accept with minor revision (please list in comments)

Comments to the Author(s)

The manuscript deals with preparation and characterization of surfactants in aqueous media. Although the characterizations are "standard" for self-assembling systems, some features and novelty makes this paper worth of publication.

I suggest revision as detailed below.

- I would recommend to add in the introduction the recent papers on novel surfactant systems and applications (see for instance J. Phys. Chem. C 2016, 120 (25), 13492-13502. <https://doi.org/10.1021/acs.jpcc.6b01282>; Colloids and Surfaces A: Physicochemical and Engineering Aspects 2015, 474, 85-91. <https://doi.org/10.1016/j.colsurfa.2015.02.037>; Appl. Sci. 2018, 8(9), 1455; <https://doi.org/10.3390/app8091455>). Introduction have to be revised.
- Tables 1 and 2, errors should be estimated and decimals provided accordingly.
- It is not clear if a_0 in Table 3 is the measured experimental parameter. Please clarify.
- Abstract should be shortened and besides the results should introduce to the thematic and propose future perspectives.

Review form: Reviewer 2

Is the manuscript scientifically sound in its present form?

No

Are the interpretations and conclusions justified by the results?

Yes

Is the language acceptable?

Yes

Is it clear how to access all supporting data?

Yes

Do you have any ethical concerns with this paper?

No

Have you any concerns about statistical analyses in this paper?

No

Recommendation?

Major revision is needed (please make suggestions in comments)

Comments to the Author(s)

Please see comments in the attached file (Appendix A).

Decision letter (RSOS-190378.R0)

15-Apr-2019

Dear Dr Liu:

Title: The Surface Adsorption, Aggregate Structure and Antibacterial Activity of Gemini Quaternary Ammonium Surfactants with Carboxylic Counterions
Manuscript ID: RSOS-190378

The editor assigned to your manuscript has now received comments from reviewers. We would like you to revise your paper in accordance with the referee and Subject Editor suggestions which can be found below (not including confidential reports to the Editor). Please note this decision does not guarantee eventual acceptance.

Please submit your revised paper before 08-May-2019. Please note that the revision deadline will expire at 00.00am on this date. If we do not hear from you within this time then it will be assumed that the paper has been withdrawn. In exceptional circumstances, extensions may be possible if agreed with the Editorial Office in advance. We do not allow multiple rounds of revision so we urge you to make every effort to fully address all of the comments at this stage. If deemed necessary by the Editors, your manuscript will be sent back to one or more of the original reviewers for assessment. If the original reviewers are not available we may invite new reviewers.

Please also include the following statements alongside the other end statements. As we cannot publish your manuscript without these end statements included, if you feel that a given heading is not relevant to your paper, please nevertheless include the heading and explicitly state that it is not relevant to your work.

- Ethics statement

Please clarify whether you received ethical approval from a local ethics committee to carry out your study. If so please include details of this, including the name of the committee that gave consent in a Research Ethics section after your main text. Please also clarify whether you received informed consent for the participants to participate in the study and state this in your Research Ethics section.

OR

Please clarify whether you obtained the necessary licences and approvals from your institutional animal ethics committee before conducting your research. Please provide details of these licences and approvals in an Animal Ethics section after your main text.

OR

Please clarify whether you obtained the appropriate permissions and licences to conduct the fieldwork detailed in your study. Please provide details of these in your methods section.

- Data accessibility

It is a condition of publication that you make available the data and research materials supporting the results in the article. Datasets should be deposited in an appropriate publicly available repository and details of the associated accession number, link or DOI to the datasets must be included in the Data Accessibility section of the article

(<http://royalsocietypublishing.org/instructions-authors#question17>). Reference(s) to datasets should also be included in the reference list of the article with DOIs (where available).

Please include a Data Availability section after your main text stating where supporting data are available from, or where they will be made available should your article be accepted for publication.

<http://datadryad.org/submit?journalID=RSOS&manu=RSOS-190378>

- Competing interests

Please include a Competing Interests section after your main text declaring any financial or non-financial competing interests. If you have no competing interests please state 'I/we have no competing interests.'

- Authors' contributions

Please include an Authors' Contributions section at the end of your main text detailing the contribution of each author. All authors should have read and approved the manuscript before submission and this should be stated in the Authors' Contributions section.

The list of Authors should meet all of the following criteria; 1) substantial contributions to conception and design, or acquisition of data, or analysis and interpretation of data; 2) drafting the article or revising it critically for important intellectual content; and 3) final approval of the version to be published.

- Acknowledgements

- Funding statement

Please include a funding section after your main text which lists the source of funding for each author.

On behalf of the Subject Editor Professor Anthony Stace and the Associate Editor Professor Kim Jelfs.

RSC Associate Editor: 1

Comments to the Author:

The authors should pay careful attention to both reviewers comments and address how they have changed the manuscript in their response letter.

Reviewers' Comments to Author:

Reviewer: 1

Comments to the Author(s)

The manuscript deals with preparation and characterization of surfactants in aqueous media. Although the characterizations are "standard" for self-assembling systems, some features and novelty makes this paper worth of publication.

I suggest revision as detailed below.

- I would recommend to add in the introduction the recent papers on novel surfactant systems and applications (see for instance J. Phys. Chem. C 2016, 120 (25), 13492–13502.

<https://doi.org/10.1021/acs.jpcc.6b01282>; Colloids and Surfaces A: Physicochemical and Engineering Aspects 2015, 474, 85–91. <https://doi.org/10.1016/j.colsurfa.2015.02.037>; Appl. Sci. 2018, 8(9), 1455; <https://doi.org/10.3390/app8091455>). Introduction have to be revised.

- Tables 1 and 2, errors should be estimated and decimals provided accordingly.

- It is not clear if a_0 in Table 3 is the measured experimental parameter. Please clarify.

- Abstract should be shortened and besides the results should introduce to the thematic and propose future perspectives.

Reviewer: 2

Comments to the Author(s)

Please see comments in the attached file.

Author's Response to Decision Letter for (RSOS-190378.R0)

See Appendix B.

RSOS-190378.R1 (Revision)

Review form: Reviewer 2

Is the manuscript scientifically sound in its present form?

Yes

Are the interpretations and conclusions justified by the results?

No

Is the language acceptable?

No

Is it clear how to access all supporting data?

Yes

Do you have any ethical concerns with this paper?

No

Have you any concerns about statistical analyses in this paper?

No

Recommendation?

Major revision is needed (please make suggestions in comments)

Comments to the Author(s)

The authors have not satisfactorily responded to following remarks:

1. "The DLS measurements and TEM visualization were done with 10-3 mol · L⁻¹ surfactants solution which is roughly two orders of magnitude higher concentration than cmc of the surfactants. Why such a high concentration was used? Knowing that the structure of aggregates can change with increasing concentration of surfactant, authors should provide data for the lower concentrations, i.e. 2 x cmc."

The authors have not explained why originally, they have chosen high surfactants' concentrations. Instead of using the opportunity to compare results obtained for low and high surfactant's concentrations they have just replaced one results with the other. The comparison should be given, as well as comment on concentration induced changes.

2. "Looking at the results of conductivity measurements Figs. S4-S6 it seems that in several cases (e.g. Fig S4 II, Fig. S5 II, Fig. S6 I and II) there is deviation from linearity at concentrations lower than cmc. This could be a consequence of ionic pairing and pre-micellar associations. I draw the attention of the authors to Zana's paper in J. Colloid Interface Sci. 246 (2002) 182-190. Such analysis of conductivity measurements should be performed."

Instead of performing analysis of conductivity data authors have just added the comment "It is

noteworthy that some results of conductivity measurements (Fig S7 II, Fig. S8 II, Fig. S9 I and II) seem to deviate from linearity at concentrations lower than CMC. This phenomenon could be a consequence of ioni paring and premicellar associations [34]."

The analysis of data should be performed, and results adequately discussed.

In addition, I would strongly suggest a correction of English language.

Decision letter (RSOS-190378.R1)

25-Jun-2019

Dear Dr Liu:

Title: The Surface Adsorption, Aggregate Structure and Antibacterial Activity of Gemini Quaternary Ammonium Surfactants with Carboxylic Counterions
Manuscript ID: RSOS-190378.R1

The editor assigned to your paper has now received comments from reviewers. We would like you to revise your paper in accordance with the referee and Subject Editor suggestions which can be found below (not including confidential reports to the Editor). Please note this decision does not guarantee eventual acceptance.

Please submit a copy of your revised paper before 18-Jul-2019. Please note that the revision deadline will expire at 00.00am on this date. If we do not hear from you within this time then it will be assumed that the paper has been withdrawn. In exceptional circumstances, extensions may be possible if agreed with the Editorial Office in advance. We do not allow multiple rounds of revision so we urge you to make every effort to fully address all of the comments at this stage. If deemed necessary by the Editors, your manuscript will be sent back to one or more of the original reviewers for assessment. If the original reviewers are not available we may invite new reviewers.

Please also include the following statements alongside the other end statements. As we cannot publish your manuscript without these end statements included, if you feel that a given heading is not relevant to your paper, please nevertheless include the heading and explicitly state that it is not relevant to your work.

- Acknowledgements

On behalf of the Subject Editor Professor Anthony Stace and the Associate Editor Professor Kim Jelfs.

RSC Associate Editor:
Comments to the Author:
(There are no comments.)

RSC Subject Editor:
Comments to the Author:
(There are no comments.)

Reviewers' Comments to Author:
Reviewer: 2

Comments to the Author(s)
The authors have not satisfactorily responded to following remarks:

1. "The DLS measurements and TEM visualization were done with 10^{-3} mol \cdot L $^{-1}$ surfactants solution which is roughly two orders of magnitude higher concentration than cmc of the surfactants. Why such a high concentration was used? Knowing that the structure of aggregates can change with increasing concentration of surfactant, authors should provide data for the lower concentrations, i.e. $2 \times$ cmc."

The authors have not explained why originally, they have chosen high surfactants' concentrations. Instead of using the opportunity to compare results obtained for low and high surfactant's concentrations they have just replaced one result with the other. The comparison should be given, as well as comment on concentration induced changes.

2. "Looking at the results of conductivity measurements Figs. S4-S6 it seems that in several cases (e.g. Fig S4 II, Fig. S5 II, Fig. S6 I and II) there is deviation from linearity at concentrations lower than cmc. This could be a consequence of ionic pairing and pre-micellar associations. I draw the

attention of the authors to Zana's paper in J. Colloid Interface Sci. 246 (2002) 182-190. Such analysis of conductivity measurements should be performed."

Instead of performing analysis of conductivity data authors have just added the comment "It is noteworthy that some results of conductivity measurements (Fig S7 II, Fig. S8 II, Fig. S9 I and II) seem to deviate from linearity at concentrations lower than CMC. This phenomenon could be a consequence of ioni paring and premicellar associations [34]."

The analysis of data should be performed, and results adequately discussed.

In addition, I would strongly suggest a correction of English language.

Author's Response to Decision Letter for (RSOS-190378.R1)

See Appendix C.

RSOS-190378.R2 (Revision)

Review form: Reviewer 2

Is the manuscript scientifically sound in its present form?

Yes

Are the interpretations and conclusions justified by the results?

Yes

Is the language acceptable?

Yes

Do you have any ethical concerns with this paper?

Yes

Have you any concerns about statistical analyses in this paper?

No

Recommendation?

Accept as is

Comments to the Author(s)

Authors have satisfactorily answered all remarks.

Decision letter (RSOS-190378.R2)

30-Jul-2019

Dear Dr Liu:

Title: The Surface Adsorption, Aggregate Structure and Antibacterial Activity of Gemini Quaternary Ammonium Surfactants with Carboxylic Counterions
Manuscript ID: RSOS-190378.R2

It is a pleasure to accept your manuscript in its current form for publication in Royal Society Open Science. The chemistry content of Royal Society Open Science is published in collaboration with the Royal Society of Chemistry.

On behalf of the Subject Editor Professor Anthony Stace and the Associate Editor Professor Kim Jelfs.

RSC Associate Editor:
Comments to the Author:
(There are no comments.)

RSC Subject Editor:
Comments to the Author:
(There are no comments.)

Reviewer(s)' Comments to Author:
Reviewer: 2

Comments to the Author(s)
Authors have satisfactorily answered all remarks.

Appendix A

Manuscript “The surface adsorption, aggregate structure and antibacterial activity of gemini quarternary ammonium surfactants with carboxylic counterions” describe the synthesis, physico-chemical and biological characterization of novel gemini surfactants. This topic is of high interest for both scientist working in the field of surfactants, but also for broader public in materials science and novel materials synthesis, especially since the concepts of green synthesis was used.

However, there are several major and minor remarks to the manuscript, in addition to English language, which need to be corrected before the paper could be published. The complete list of remarks is given below.

Major remarks

- a) The first step in physico-chemical characterisation of any novel surfactant is determination of its Krafft temperature, to determine the temperature range in which it can form aggregates. This data is missing in this manuscript and should be added.
- b) The DLS measurements and TEM visualization were done with 10^{-3} mol L⁻¹ surfactants solution which is roughly two orders of magnitude higher concentration than cmc of the surfactants. Why such a high concentration was used? Knowing that the structure of aggregates can change with increasing concentration of surfactant authors should provide data for the lower concentrations, i.e. 2 x cmc.
- c) In the description of Gibbs adsorption equation there are several points that need to be addressed:
 - The reference given as a source from which equation is taken is incorrect. M.J. Rosen, *Surfactants and Interfacial Phenomena*, second ed., Wiley, New York, 1989. Is one of the correct references.
 - In equation 1 there is no need for the middle part. Also please take notice that physical variables should be written italic. The correction should be made through the entire text.
 - Description of prefactor n should be extended and more precise. A reference Z.X. Li, C.C. Dong, R.K. Thomas, *Langmuir* 15 (1999) 4392, should be added.
- d) The reference for equation for Gibbs energy of adsorption (eq. 4) and Gibbs energy of micellization (eq. 3) is also incorrect, it is reference of a review of M.J. Rosen, *Surfactants and Interfacial Phenomena* book, not the book itself. The Gibbs energy of micellization should be calculated using equation:

$$\Delta G_M^0 = RT(\frac{1}{2} + \beta) \ln \text{cmc} - (RT/2) \ln 2$$

given in Zana, *Langmuir* 12 (1996) 1208, using data from conductivity measurements. In line with this, all data should be recalculated and data from Tables 1 and 2 combined into one table, for better clarity. In the eq. 4 the factor 6.022, should be corrected to 6.023, according to Rosen.

Also equation 5 for is not from the reference 18. Above Zana's equation should be used and data recalculated.

- e) The discussion of properties synthesised surfactants can be improved by comparison with the behaviour of conventional quarternary ammonium surfactants with carboxylate anion, although the hydrophobic chain lengths are different, i.e. C12 and C14 (Yan et al. *J. Surf. Deterg.* 15 82012) 593.).

- f) Looking at the results of conductivity measurements Figs. S4-S6 it seems that in several cases (e.g. Fig S4 II, Fig. S5 II, Fig. S6 I and II) there is deviation from linearity at concentrations lower than cmc. This could be a consequence of ionic pairing and pre-micellar associations. I draw attention of the authors to Zana's paper in J. Colloid Interface Sci. 246 (2002) 182-190. Such analysis of conductivity measurements should be performed.

Minor remarks

- a) Page 2, line 10 Hofmeister should be written with capital H
- b) The shorten notation for synthesised surfactants used was $n\text{-N}_2\text{N-}n\text{-2Y}$. It would be simpler and easier for the readers from the field to change it to commonly used $n\text{-2-}n\text{-2Y}$.
- c) Page 4, line 7, "to find" should be changed to "to determine"
- d) The notations should be uniform throughout the paper

Appendix B

Dear Dr Laura Smith:

Our manuscript–The Surface Adsorption, Aggregate Structure and Antibacterial Activity of Gemini Quaternary Ammonium Surfactants with Carboxylic Counterions (RSOS-190378) has been revised according to your comments and the itemized response to your comments is attached. I am very happy that you give us the chance to revise my manuscript and very sorry to bring you so much trouble because of our careless in the manuscript.

Thanks very much again for your attention to our paper. Once again, thank you for your help to our paper processing.

According to the reviewer comments, we have made the revisions as follows:

The comment from Reviewer 1:

Comment 1

I would recommend to add in the introduction the recent papers on novel surfactant systems and applications (see for instance J. Phys. Chem. C 2016, 120 (25), 13492-13502. <https://doi.org/10.1021/acs.jpcc.6b01282>; Colloids and Surfaces A: Physicochemical and Engineering Aspects 2015, 474, 85-91. <https://doi.org/10.1016/j.colsurfa.2015.02.037>; Appl. Sci. 2018, 8(9), 1455; <https://doi.org/10.3390/app8091455>). Introduction have to be revised.

Answers and Revisions:

In the manuscript, introduction have been revised. The recent papers on novel surfactant systems and applications also have been added.

Comment 2

Tables 1 and 2, errors should be estimated and decimals provided accordingly.

Answers and Revisions:

In the manuscript, the data from Tables 1 and 2 has been combined into one table, for better clarity, and the errors had been provided accordingly.

Comment 3

It is not clear if a_0 in Table 3 is the measured experimental parameter. Please clarify.

Answers and Revisions:

In the manuscript, a_0 was explained as follows:

where a_0 can be approximated to a half of A_{min} with $n=2$ [40].

Comment 4

Abstract should be shortened and besides the results should introduce to the thematic and propose future perspectives.

Answers and Revisions:

Abstract is revised as follows:

A group of gemini quaternary ammonium surfactants with the formula $C_nH_{2n+1}CONH(CH_2)_2N^+(CH_3)_2(CH_2)_2N^+(CH_3)_2(CH_2)_2NHCOC_nH_{2n+1}\cdot 2Y$ ($n=11, 13$ and 15 , $Y=HCOO^-$, CH_3COO^- and $CH_3CHOHCOO^-$) have been synthesized by a counterion conversion process and characterized by Fourier transform infrared spectroscopy and mass spectroscopy. Their adsorption and self-aggregation properties are investigated by surface tension, conductivity, dynamic light scattering and transmission electron microscopy measurements (TEM). The results show that these synthesized surfactants reduce the surface tension of water to a minimum value of $26.51 \text{ mN}\cdot\text{m}^{-1}$ at a concentration of $5.72 \times 10^{-2} \text{ mmol}\cdot\text{L}^{-1}$. Furthermore, the increased alkyl chain length of the carboxylic counterions leads to the increased critical micelle concentration (CMC), the decreased degree of counterion binding (β), and the decreased self-assembly tendency, but the minimum area per surfactant molecule (A_{min}) adsorbed at the air-aqueous solution are similar. TEM images reveal that these surfactants self-assemble spontaneously into aggregates with vesicles or bilayers structure. It is also found that they have superior antibacterial activity at a concentration of $0.1 \text{ g}\cdot\text{L}^{-1}$. The high surface activity and high antibacterial activity of these gemini surfactants containing different carboxylic counterions bring more possibilities for their application in the field of biomedicine.

The comment from Reviewer 2:

Major remarks

Comment a:

The first step in physico-chemical characterisation of any novel surfactant is determination of its Krafft temperature, to determine the temperature range in which it can form aggregates. This data is missing in this manuscript and should be added.

Answers and Revisions:

The krafft temperature of these surfactants has been added as follows:

2.4. Measurement of Krafft Temperature (T_k)

1% aqueous solution of the surfactant were prepared and kept in refrigerator for 10 h at about 7 °C. Then the precipitated system was taken out of the refrigerator and placed in a low-temperature thermostat bath. DDS-307 conductivity meter (cell constant is 0.997 cm⁻¹) was used to perform the conductivity measurements of these surfactants solution and each temperature gradient was measured three times to determine the average value with the standard deviation was less than 0.2 μS·cm⁻¹. The initial temperature of the solution was controlled as 7±0.1°C and the measurements were carried out in 1 °C increment with constant stirring by a glass rod.

3.2. Krafft Temperature (T_k)

The krafft temperature of these surfactants was determined from the plot of the specific conductance (κ) vs. temperature in aqueous solution. The corresponding graphs have been shown in Figure S4-6. Seen from Figure S4-6, in the relatively low temperature range, the conductance of the surfactant solution has a small change with the increase of temperature, and then the sharp increase is observed within a relatively narrow temperature range that represents the sharp increase in surfactant solubility. The T_k of the surfactants can be obtained from the inflection point of the κ -T curve (Table 1).

Table 1 The T_k of 11-2-11-2Y, 13-2-13-2Y, 15-2-15-2Y (Y=HCOO⁻, CH₃COO⁻ and CH₃CHOHCOO⁻) aqueous solution

gemini surfactants	T_k (°C)
11-2-11-2HCOO ⁻	17
11-2-11-2CH ₃ COO ⁻	18
11-2-11-2CH ₃ CHOHCOO ⁻	18
13-2-13-2HCOO ⁻	18
13-2-13-2CH ₃ COO ⁻	18
13-2-13-2CH ₃ CHOHCOO ⁻	17
15-2-15-2HCOO ⁻	18
15-2-15-2CH ₃ COO ⁻	18
15-2-15-2CH ₃ CHOHCOO ⁻	19

Comment b:

The DLS measurements and TEM visualization were done with 10⁻³ mol·L⁻¹ surfactants solution which is roughly two orders of magnitude higher concentration than cmc of the surfactants. Why such a high concentration was used? Knowing that the structure of aggregates can change with increasing concentration of surfactant, authors should provide data for the lower concentrations, i.e. 2 x cmc.

Answers and Revisions:

Measurement of DLS and TEM are done with the concentration of $2\times\text{CMC}$.

The revision is as follows:

3.5. DLS and TEM Measurements

R_h distributions of these synthesized gemini quaternary ammonium surfactants are between 40 nm and 2000 nm with little distinction, as shown in Figure S10. In the case of the same hydrophobic chain length, two peaks are shown in the surfactants with the formate counterions. One of that is distributed in tens of nanometers though the intensity of the peak of 13-2-13-2HCOO⁻ is obviously lower than others. Beyond that, the peak values of gemini surfactants with acetate or lactate counterions are generally distributed in hundreds of nanometers. The result indicates that the surfactants with formate counterions tend to form smaller aggregates. Furthermore, the peak of tens of nanometers may correspond to micro vesicles, and the peak of hundreds of nanometers may be assigned to more complex aggregates, such as multilamellar vesicles, tubular micelle, lamellar micelle, etc. Accordingly, the morphology of aggregates of these gemini surfactants in aqueous solution of the concentration of $2\times\text{CMC}$ are further carried out on TEM measurement, which are shown in Figure 3-5.

Seen from Figure 3-5, the size of these aggregates is practically consistent with the DLS results. In the system of 11-2-11-2Y, the vesicles structure is seen in the samples with formate and acetate counterions, but the aggregate of the latter is deformed and presents an elliptic structure. However, the sample with lactate counterion shows a mass of bilayers micelles. This indicates that, at the low concentration of these gemini surfactant solution, the micelles show a trend of transition from vesicles to bilayers aggregates with the increasing alkyl chain length of counterions. This trend is also present in the system of 13-2-13-2Y, although the bilayers structure has already been observed in the sample with formate counterion. Based on the conductivity data, β values of these surfactants with lactate counterions are smaller, which indicates the lower concentration of counterions in the interface between the micelle and the aqueous solution. This would result in the compression of the diffusion bilayer and smaller cross-sectional area of the micelle-aqueous interface, when the surfactants are closely aligned. Therefore, in the similar size of micelles, these surfactants with longer alkyl chain length of counterions tend to form flat structures. However, in the system of 15-2-15-2Y, the aggregates with vesicles structure are formed, and the carboxylic counterions seem have little effect on the structure of the aggregates. The formation of various micelles is mainly due to the self-organization of

quaternary ammonium cation driven by entropy. So one of the major factors leading to the differences is the distinction of hydrophobic chain length of these systems.

Figure 3. TEM micrographs of 11-2-11-2Y in aqueous solution of the concentration of $2\times\text{CMC}$, I: $\text{Y} = \text{HCOO}^-$, II: CH_3COO^- , and III: $\text{CH}_3\text{CHOHCOO}^-$

Figure 4. TEM micrographs of 13-2-13-2Y in aqueous solution of the concentration of $2\times\text{CMC}$, I: $\text{Y} = \text{HCOO}^-$, II: CH_3COO^- , and III: $\text{CH}_3\text{CHOHCOO}^-$

Figure 5. TEM micrographs of 15-2-15-2Y in aqueous solution of the concentration of $2\times\text{CMC}$, I: $\text{Y} = \text{HCOO}^-$, II: CH_3COO^- , and III: $\text{CH}_3\text{CHOHCOO}^-$

Comment c:

In the description of Gibbs adsorption equation there are several points that need to be addressed:

- The reference given as a source from which equation is taken is incorrect. M.J. Rosen, *Surfactants and Interfacial Phenomena*, second ed., Wiley, New York, 1989. Is one of the correct references.

- In equation 1 there is no need for the middle part. Also please take notice that physical variables should be written italic. The correction should be made through the entire text.

- Description of prefactor n should be extended and more precise. A reference Z.X. Li, C.C. Dong, R.K. Thomas, *Langmuir* 15 (1999) 4392, should be added.

Answers and Revisions:

- The reference- M.J. Rosen, *Surfactants and Interfacial Phenomena*, second ed., Wiley, New York, 1989 has been added.

- Equation 1 has been revised as follows:

$$\Gamma_{max} = \left(\frac{-1}{2.303nRT} \right) \left(\frac{d\gamma}{d \log c} \right) \quad (1)$$

Physical variables has been written italic through the entire text.

- Description of prefactor n is revised as follows:

The value of n has no effect on the variation trend of the values of Γ_{max} and A_{min} . However, it has been widely assumed that it is more appropriate to use a value of 2 for n , because some reports suggest that as n is 3, the calculated A_{min} is unacceptably large [26]. In this case, a value of 2 for n is used to calculate.

26. Li ZX, Dong CC, Thomas RK. 1999 Neutron Reflectivity Studies of the Surface Excess of Gemini Surfactants at the Air-Water Interface. *Langmuir* **15**, 4392-4396. (doi: 10.1021/la981551u)

Comment d:

The reference for equation for Gibbs energy of adsorption (eq. 4) and Gibbs energy of micellization (eq. 3) is also incorrect, it is reference of a review of M.J. Rosen, Surfactants and Interfacial Phenomena book, not the book itself. The Gibbs energy of micellization should be calculated using equation:

$$\Delta G_M^\circ = RT(\frac{1}{2} + \beta) \ln cmc - (RT/2) \ln 2$$

given in Zana, *Langmuir* 12 (1996) 1208, using data from conductivity measurements. In line with this, all data should be recalculated and data from Tables 1 and 2 combined into one table, for better clarity. In the eq. 4 the factor 6.022, should be corrected to 6.023, according to Rosen.

Also equation 5 for is not from the reference 18. Above Zana's equation should be used and data recalculated.

Answers and Revisions:

The reference for equation for Gibbs energy of adsorption (eq. 4) and Gibbs energy of micellization (eq. 3) is revised as follows:

27. Desnoyers JE. 1992 Surfactants and interfacial phenomena 2nd edition. By Milton J. Rosen, John Wiley and Sons, New York, 1989. 431 pp. *J. Colloid Interf. Sci.* **149** (1), 299-300. (doi: 10.1016/0021-9797(92)90419-M)

Equation (4) has been revised as follows:

$$\Delta G_{ads} = \Delta G_m^\theta - 6.023(\gamma_0 - \gamma_{CMC})A_{min} \quad (4)$$

Equation (5) has been revised as follows:

ΔG_m^θ was calculated according to the following equation [36],

$$\Delta G_m^\theta = RT(\frac{1}{2} + \beta) \ln CMC - \frac{1}{2} RT \ln 2 \quad (5)$$

36. Zana R. 1996 Critical micellization concentration of surfactants in aqueous solution and free energy of micellization. *Langmuir* **12**, 1208-1211. (doi: 10.1021/la950691q)

All data has been recalculated and data from Tables 1 and 2 has combined into one table. The revision is as follows:

Table 2 Characteristic parameters of 11-2-11-2Y, 13-2-13-2Y, 15-2-15-2Y (Y=HCOO⁻, CH₃COO⁻, and CH₃CHOHCOO⁻) aqueous solution at 25 °C

gemini surfactant	CMC (mmol·L ⁻¹)		γ_{CMC} (mN·m ⁻¹)	pC ₂₀	Γ_{max} (×10 ⁻¹⁰ mol·cm ⁻²)	A_{min} (nm ²)	ΔG_{ads} (kJ·mol ⁻¹)	ΔG_m^{θ} (kJ·mol ⁻¹)		β
	method 1 ^a	method 2 ^b						method 1 ^a	method 2 ^b	
11-2-11-2HCOO ⁻	5.72×10 ⁻²	5.73×10 ⁻²	26.51	5.42	2.09	0.79	-55.81	-34.17	-26.43	0.556
11-2-11-2CH ₃ COO ⁻	6.32×10 ⁻²	6.18×10 ⁻²	26.92	5.37	1.91	0.87	-57.54	-33.92	-26.06	0.549
11-2-11-2CH ₃ CHOHCOO ⁻	0.101	0.105	27.63	5.14	1.90	0.88	-56.28	-32.76	-23.84	0.512
13-2-13-2HCOO ⁻	3.16×10 ⁻²	3.18×10 ⁻²	27.62	5.40	2.41	0.69	-54.08	-35.64	-26.76	0.509
13-2-13-2CH ₃ COO ⁻	4.09×10 ⁻²	4.10×10 ⁻²	27.99	5.20	2.69	0.62	-51.43	-35.00	-25.70	0.492
13-2-13-2CH ₃ CHOHCOO ⁻	8.92×10 ⁻²	6.78×10 ⁻²	28.49	4.87	2.54	0.65	-50.10	-33.07	-23.53	0.453
15-2-15-2HCOO ⁻	1.05×10 ⁻²	1.41×10 ⁻²	28.32	5.61	3.35	0.50	-51.52	-38.37	-27.49	0.462
15-2-15-2CH ₃ COO ⁻	2.09×10 ⁻²	2.19×10 ⁻²	28.35	5.34	3.14	0.53	-50.60	-36.67	-25.86	0.440
15-2-15-2CH ₃ CHOHCOO ⁻	2.63×10 ⁻²	3.34×10 ⁻²	29.09	5.23	3.09	0.54	-50.06	-36.10	-24.93	0.442

The method 1 stands for surface tension measurement. The method 2 stands for conductivity measurement. The standard error σ_{μ} are $\sigma_{\mu}(\text{CMC}) = 1 \times 10^{-3} \text{ mmol} \cdot \text{L}^{-1}$, $\sigma_{\mu}(\gamma_{CMC}) = 0.01 \text{ mN} \cdot \text{m}^{-1}$, $\sigma_{\mu}(\text{pC}_{20}) = 0.01$, $\sigma_{\mu}(\Gamma_{max}) = 0.01 \times 10^{-10} \text{ mol} \cdot \text{cm}^{-2}$, $\sigma_{\mu}(A_{min}) = 0.01 \text{ nm}^2$, $\sigma_{\mu}(A_{min}) = 0.01 \text{ nm}^2$, $\sigma_{\mu}(A_{min}) = 0.01 \text{ nm}^2$, $\sigma_{\mu}(\Delta G_{ads}) = -1 \text{ kJ} \cdot \text{mol}^{-1}$, $\sigma_{\mu}(\Delta G_{ads}) = -1 \text{ kJ} \cdot \text{mol}^{-1}$.

Comment e:

The discussion of properties synthesised surfactants can be improved by comparison with the behaviour of conventional quarternary ammonium surfactants with carboxylate anion, although the hydrophobic chain lengths are different, i.e. C12 and C14 (Yan et al. *J. Surf. Deterg.* 15 82012) 593.).

Answers and Revisions:

The discussion has been added as follows:

The CMC values of the corresponding conventional quarternary ammonium surfactants with carboxylate anion $[\text{CH}_3(\text{CH}_2)_n\text{CH}_2\text{N}^+(\text{CH}_3)_3\cdot\text{A}^-, n=10, 12, \text{A}=\text{HCOO}^-, \text{CH}_3\text{COO}^-]$ are 2 order of magnitude higher than the synthesised gemini surfactants, although the hydrophobic chain lengths are different [29].

29. Yan HC, Li QX, Geng T, Jiang YJ. 2012 Properties of the Quaternary Ammonium Salts with Novel Counterions. *J. Surf. Deterg.* **15**, 593-599. (doi: 10.1007/s11743-012-1347-y)

Comment f:

Looking at the results of conductivity measurements Figs. S4-S6 it seems that in several cases (e.g. Fig S4 II, Fig. S5 II, Fig. S6 I and II) there is deviation from linearity at concentrations lower than cmc. This could be a consequence of ioni paring and premicellar associations. I draw attention of the authors to Zana's paper in *J. Colloid Interface Sci.* 246 (2002) 182-190. Such analysis of conductivity measurements should be performed.

Answers and Revisions:

The discussion has been added as follows:

It is noteworthy that some results of conductivity measurements (Fig S7 II, Fig. S8 II, Fig. S9 I and II) seem to deviate from linearity at concentrations lower than CMC. This phenomenon could be a consequence of ioni paring and premicellar associations [34].

34. Zana R. 2002 Alkanediyl- α,ω -bis(dimethylalkylammonium bromide) Surfactants: 10. Behavior in Aqueous Solution at Concentrations below the Critical Micellization Concentration: An Electrical Conductivity Study. *J. Colloid Interface Sci.* **246**, 182-190. (doi: 10.1006/jcis.2001.7921)

Minor remarks**Comment a:**

Page 2, line 10 Hofmeister should be written with capital H.

Answers and Revisions:

In the manuscript, Hofmeister has been written with capital H.

Comment b:

The shorten notation for synthesised surfactants used was $n\text{-N}2\text{N-}n\text{-}2\text{Y}$. It would be simpler and easier for the readers from the field to change it to commonly used $n\text{-}2\text{-}n\text{-}2\text{Y}$.

Answers and Revisions:

In the manuscript, the shorten notation for synthesised surfactants has been changed to $n\text{-}2\text{-}n\text{-}2\text{Y}$.

Comment c:

Page 4, line 7, “to find” should be changed to “to determine”.

Answers and Revisions:

In the manuscript, “to find” has been changed to “to determine”.

Comment d:

The notations should be uniform throughout the paper.

Answers and Revisions:

The notations has been uniform throughout the paper.

At last, we are appreciated for your assistance and we hope our revised paper could be published in the journal.

Best wishes,

Sincerely yours,

Hongqin Liu

Appendix C

Dear Dr Laura Smith:

Our manuscript–The Surface Adsorption, Aggregate Structure and Antibacterial Activity of Gemini Quaternary Ammonium Surfactants with Carboxylic Counterions (RSOS-190378R1) has been revised according to your comments and the itemized response to your comments is attached. I am very happy that you give us the chance to revise my manuscript and very sorry to bring you so much trouble because of our careless in the manuscript.

Thanks very much again for your attention to our paper. Once again, thank you for your help to our paper processing.

According to the reviewer comments, we have made the revisions as follows:

The comment from Reviewer 2:

Comment 1:

“The DLS measurements and TEM visualization were done with 10^{-3} mol·L⁻¹ surfactants solution which is roughly two orders of magnitude higher concentration than cmc of the surfactants. Why such a high concentration was used? Knowing that the structure of aggregates can change with increasing concentration of surfactant, authors should provide data for the lower concentrations, i.e. 2 x cmc.”

The authors have not explained why originally, they have chosen high surfactants' concentrations. Instead of using the opportunity to compare results obtained for low and high surfactant's concentrations they have just replaced one results with the other. The comparison should be given, as well as comment on concentration induced changes.

Answers and Revisions:

Measurement of DLS and TEM are done at the concentration of $2 \times \text{CMC}$ and 1×10^{-3} mol·L⁻¹.

The revision is as follows:

The DLS data and TEM micrographs of surfactant solutions at a concentration of $2 \times \text{CMC}$ are frequently used to evaluate their molecular self-assembly behavior. In the antibacterial activity subsequently discussed, at this concentration (about 0.05 g·L⁻¹), the bacteriostatic efficiency of these gemini surfactants aqueous solution was

53.9%-80.2% for Gram-positive bacteria and Gram-negative bacteria. However, at the concentration of $1 \times 10^{-3} \text{ mol} \cdot \text{L}^{-1}$ (about $0.1 \text{ g} \cdot \text{L}^{-1}$), the antibacterial efficiency of these surfactant solutions is all above 75.3%, most of which are 80%-95%. So, the self-assembly behavior of these gemini surfactants aqueous solution at higher concentration ($1 \times 10^{-3} \text{ mol} \cdot \text{L}^{-1}$) was also investigated.

Seen from Figure S5, the R_h values of these surfactant solutions with the concentration of $2 \times \text{CMC}$ are distributed in the range of 40-2000 nm, while the R_h values of $1 \times 10^{-3} \text{ mol} \cdot \text{L}^{-1}$ are distributed in the range of 10-3000 nm. It is suggested that the increasing concentration of these surfactant solutions leads to more diverse and larger size aggregates. At the concentration of $2 \times \text{CMC}$, two peaks are shown in the surfactants with formate counterion in the case of the same hydrophobic chain length. One of that is distributed in tens of nanometers though the intensity of the peak of 13-2-13-2HCOO⁻ is not obvious and the other are generally distributed in hundreds of nanometers. The result indicates that the surfactants with formate counterion tend to form smaller aggregates at the low concentration. At the concentration of $1 \times 10^{-3} \text{ mol} \cdot \text{L}^{-1}$, at least two hydrated radius distribution peaks were observed. The peak of tens of nanometers may correspond to micro vesicles, and the peak of hundreds of nanometers may be assigned to more complex aggregates, such as multilamellar vesicles, tubular micelle, lamellar micelle, etc. Accordingly, the morphology of aggregates of these gemini surfactants in aqueous solution with the concentration of $2 \times \text{CMC}$ and $1 \times 10^{-3} \text{ mol} \cdot \text{L}^{-1}$ have been carried out on TEM measurement, which are shown in Figure 3-5.

Seen from Figure 3-5, the size of these aggregates is practically consistent with the results of DLS. In the series of 11-2-11-2Y, at the concentration of $2 \times \text{CMC}$, vesicles with a diameter of about 300 nm are formed in aqueous solution of 11-2-11-2HCOO⁻ and 11-2-11-2CH₃COO⁻ (Figure 3 A-I, A-II), while at the concentration of $1 \times 10^{-3} \text{ mol} \cdot \text{L}^{-1}$, multilamellar vesicles (MLV) with a diameter of about 500 nm are formed (Figure 3 B-I, B-II). In addition, at the concentration of $2 \times \text{CMC}$, a mass of hollow slender bilayers micelles are observed in aqueous solution of 11-2-11-2CH₃CHOHCOO⁻ (Figure 3 A-III), while at the concentration of $1 \times 10^{-3} \text{ mol} \cdot \text{L}^{-1}$, vesicles are formed (Figure 3 B-III). This indicates that the introduction of -OH and the increase in the length of the alkyl carbon chain in the counterions cause the deformation of the aggregates at the concentration of $2 \times \text{CMC}$, but the effect of counterions seems to be insignificant at relatively high concentration. This trend is

also present in the series of 13-2-13-2Y. The mixed micelles of vesicles and deformed bilayers structure are observed in Figure 4 A-I, while uniform short tubular micelles are observed in Figure 4 A-II and A-III. However, hollow and longish tubular micelles are all observed among that at a high concentration (Figure 4 B-I, B-II, B-III). In spite of that, in the series of 15-2-15-2Y, the counterions have little influence on the self-assembly behavior of these surfactants whether at a low or high concentration. Seen from Figure 5, the vesicles with a diameter of about 100-200 nm are formed at the concentration of $2\times\text{CMC}$, while a mixed micelles of vesicle and longish tubular structure are formed at the concentration of $1\times 10^{-3} \text{ mol}\cdot\text{L}^{-1}$. The formation of various micelles is mainly due to the self-organization of quaternary ammonium cation driven by entropy. So, one of the major factors leading to the differences may be the distinction of hydrophobic chain length of these systems.

Figure S5 R_h distribution of n -2- n -2Y, A: $2\times\text{CMC}$, B: $1\text{E-}3\text{mol}\cdot\text{L}^{-1}$, I: $n=11$, II: $n=13$, III: $n=15$, Y= HCOO^- (■), CH_3COO^- (●), and $\text{CH}_3\text{CHOHCOO}^-$ (▲) in aqueous

solution

Figure 3. TEM micrographs of 11-2-11-2Y in aqueous solution of the concentration of $2 \times \text{CMC}$ (A) and $1 \times 10^{-3} \text{ mol} \cdot \text{L}^{-1}$ (B), I: $\text{Y} = \text{HCOO}^-$, II: CH_3COO^- , and III: $\text{CH}_3\text{CHOHCOO}^-$

Figure 4. TEM micrographs of 13-2-13-2Y in aqueous solution of the concentration of $2\times\text{CMC}$ (A) and $1\times 10^{-3}\text{ mol}\cdot\text{L}^{-1}$ (B), I: $\text{Y} = \text{HCOO}^-$, II: CH_3COO^- , and III: $\text{CH}_3\text{CHOHCOO}^-$

Figure 5. TEM micrographs of 15-2-15-2Y in aqueous solution of the concentration of $2\times\text{CMC}$ (A) and $1\times 10^{-3} \text{ mol}\cdot\text{L}^{-1}$ (B), I: $\text{Y} = \text{HCOO}^-$, II: CH_3COO^- , and III: $\text{CH}_3\text{CHOHCOO}^-$

Comment 2:

“Looking at the results of conductivity measurements Figs. S4-S6 it seems that in several cases (e.g. Fig S4 II, Fig. S5 II, Fig. S6 I and II) there is deviation from linearity at concentrations lower than cmc. This could be a consequence of ioni paring and premicellar associations. I draw the attention of the authors to Zana’s paper in J. Colloid Interface Sci. 246 (2002) 182-190. Such analysis of conductivity measurements should be performed.”

Instead of performing analysis of conductivity data authors have just added the comment “It is noteworthy that some results of conductivity measurements (Fig S7 II, Fig. S8 II, Fig. S9 I and II) seem to deviate from linearity at concentrations lower than CMC. This phenomenon could be a consequence of ioni paring and premicellar associations [34].”

The analysis of data should be performed, and results adequately discussed.

Answers and Revisions:

The analysis of data has been performed, and the discussion has been revised as follows:

This phenomenon could be a consequence of ioni paring or premicellar associations, which can be analyzed by the shape of κ vs c and molar conductivity Λ ($\Lambda = (\kappa - \kappa_0) / c$, κ_0 is the conductivity of water) vs $c^{0.5}$ plots at $c < \text{CMC}$ [34]. Ion pairing results in a neutralization of electrical charges and thus in a loss of conductivity, so the corresponding κ vs c and Λ vs $c^{0.5}$ plots curve toward the

concentration. On the other hand, premicellar association of ionic surfactant gives rise to small aggregates of surfactant ions, which results in an increase of conductivity of the solution, therefore the corresponding κ vs c plot will show a curvature toward the κ axis. In addition, Λ will increase with c . At higher concentration, where micelles appear in the solution and bind counterions, the value of Λ will start decreasing. Thus, the plot of Λ vs $c^{0.5}$ will present a maximum. So according to Figure S3 A-II, B-II, C-I and C-II, and Figure S4, ion pairing may occur in solutions of 11-2-11-2CH₃COO⁻, 13-2-13-2CH₃COO⁻ and 15-2-15-2CH₃COO⁻, and premicellar association may be present for 15-2-15-2HCOO⁻.

Figure S4 Plots of $10^2\Lambda$ vs $c^{0.5}$ for 11-2-11-2CH₃COO⁻ (I), 13-2-13-2CH₃COO⁻ (I), 15-2-15-2HCOO⁻ (III), and 15-2-15-2CH₃COO⁻ (IV) at 25 °C

In addition, in the manuscript we have corrected English.

At last, we are appreciated for your assistance and we hope our revised paper could be published in the journal.

Best wishes,

Sincerely yours,

Hongqin Liu